# Acute Caffeine Intake Enhances Mean Power Output and Bar Velocity during the Bench Press Throw in Athletes Habituated to Caffeine

**DOI:** 10.3390/nu12020406

**Published:** 2020-02-04

**Authors:** Michal Wilk, Aleksandra Filip, Michal Krzysztofik, Mariola Gepfert, Adam Zajac, Juan Del Coso

**Affiliations:** 1Institute of Sport Sciences, The Jerzy Kukuczka Academy of Physical Education, 40-065 Katowice, Poland; a.filip@awf.katowice.pl (A.F.); m.krzysztofik@awf.katowice.pl (M.K.); m.gepfert@awf.katowice.pl (M.G.); a.zajac@awf.katowice.pl (A.Z.); 2Centre for Sport Studies, Rey Juan Carlos University, 28942 Fuenlabrada, Spain; juan.delcoso@urjc.es

**Keywords:** ballistic exercise, upper limbs, resistance exercise, ergogenic substances, sport performance

## Abstract

Background: The main objective of the current investigation was to evaluate the effects of caffeine on power output and bar velocity during an explosive bench press throw in athletes habituated to caffeine. Methods: Twelve resistance trained individuals habituated to caffeine ingestion participated in a randomized double-blind experimental design. Each participant performed three identical experimental sessions 60 min after the intake of a placebo, 3, and 6 mg/kg/b.m. of caffeine. In each experimental session, the participants performed 5 sets of 2 repetitions of the bench press throw (with a load equivalent to 30% repetition maximum (RM), measured in a familiarization trial) on a Smith machine, while bar velocity and power output were registered with a rotatory encoder. Results: In comparison to the placebo, the intake of caffeine increased mean bar velocity during 5 sets of the bench press throw (1.37 ± 0.05 vs. 1.41 ± 0.05 and 1.41 ± 0.06 m/s for placebo, 3, and 6 mg/kg/b.m., respectively; *p* < 0.01), as well as mean power output (545 ± 117 vs. 562 ± 118 and 560 ± 107 W; *p* < 0.01). However, caffeine was not effective at increasing peak velocity (*p* = 0.09) nor peak power output (*p* = 0.07) during the explosive exercise. Conclusion: The acute doses of caffeine before resistance exercise may increase mean power output and mean bar velocity during the bench press throw training session in a group of habitual caffeine users. Thus, caffeine prior to ballistic exercises enhances performance during a power-specific resistance training session.

## 1. Introduction

Caffeine (CAF) is one of the most common substances used in sport which enhances physical performance [1]. Although CAF may affect various body tissues [2,3], there is a growing body of evidence in animal [4] and human models [5] sustaining the ability of CAF to act as an adenosine antagonist, as the main mechanism behind CAF ergogenic effect. Current research recommends doses of CAF ranging from 3 to 6 mg/kg body mass to elicit ergogenic benefits, while the time of ingestion (from 30 to 90 min before exercise) and the form of ingestion (pills, liquid, chewing gum) are less significant as CAF is rapidly absorbed after ingestion [6]. However, the optimal protocol of CAF supplementation may differ based on the type and duration of exercise, previous habituation to CAF and the type of muscle contraction [6,7,8,9,10].

Acute CAF intake causes a slightly different response to upper and lower body exercise [10], despite the lack of a mechanisms explaining this phenomenon. A recent meta-analysis about the effects of CAF on muscle strength and power output found that CAF significantly improved upper but not lower body strength [7]. Although the outcomes of the first investigations were contradictory [11,12,13], more recent studies have found a clear effect of CAF on several forms of upper body muscle performance [14,15,16], when the used dose was at least 3 mg/kg/b.m. [17]. Interestingly, the effect of CAF on upper body muscle performance may be partially dampened in athletes habituated to CAF because the regular use of CAF-containing products [18] may develop tolerance to this substance. It must be noticed that this decreased ergogenic effect of CAF is not removed even with the ingestion of doses >9 mg/kg/b.m. [19,20].

Most studies on the acute effects of CAF on upper body muscle performance used the bench press exercise. The bench press exercise is widely used as a means of developing strength and power of the upper limbs [21,22]. However, other authors, recommend the use of ballistic exercises for the development of upper body power, such as the bench press throw (BPT) [23,24]. Specifically, to increase power output, the loads ranging from 0% to 50% of one-repetition maximum (1 RM) moved at maximal speed are recommended as the most potent loading stimuli for improving power output [23]. However, this type of routine can only be performed on a Smith machine while using the BPT exercise (i.e., maximal bar velocity is obtained at the moment of throw). The traditional free-weight bench press exercise does not allow for the attainment of maximal velocity of execution (i.e., velocity equals 0 m/s at maximal arm extension). Thus, ballistic exercises could be an optimal choice for power training as they allow for greater velocity, and muscle activation in comparison to similar traditional resistance training routines [21]. Perhaps, the main asset of the BPT is the maximum acceleration of a given load, which ultimately produces high movement velocity in a short time [25]. In this respect, the loads applied in ballistic exercises during training will depend on the specific requirements of particular sport disciplines and will determine success in numerous power-based competitions [23]. Furthermore, the BPT performance has been associated with overall performance in different sport-specific tasks [26,27,28,29]. Therefore, it seems reasonable to use the BPT as a means of testing upper-body ballistic performance. Although the BPT is indicated as the most effective exercise for developing power of the upper limbs [30], previous studies have not determined the acute effects of CAF on power output and bar velocity during this type of exercise.

Burke [31] has suggested that, in the current literature, there is a lack of data about the practical use of supplements in competitive sports because experimental protocols are often different from sports practice. In case of CAF, most studies considering the acute effects of this supplement were assessed on the basis of only a single set of an exercise [7,8,32], while real resistance training sessions in trained individuals rarely contain only one set of a particular exercise [31]. On the other hand, the investigations analyzing the acute effects of CAF on performance during successive sets of resistance exercises are scarce. Bowtell et al. [3] showed that pre-exercise CAF (6 mg/kg/b.m.) intake improved total exercise time during 5 sets of one-legged knee extensions performed to failure in comparison to a placebo (PLAC). It is worth noting that this ergogenic effect was achieved despite significantly lower muscle phosphocreatine concentration (PCr) and pH in the latter sets of an exercise in the CAF trial. Further, CAF ingestion (6 mg/kg/b.m.) attenuated the increase in interstitial potassium during one-legged knee extensions at 20 W (10 min) and 50 W (3 × 3 min) measured using microdialysis [33], which resulted in a 16% improvement in high-intensity exercise capacity.

Most studies related to the acute impact of CAF intake on power output and bar velocity have used participants unhabituated to CAF or individuals with low-to-moderate daily consumption of this stimulant [11,16,34]. However, the analysis of urinary CAF concentrations after official competitions suggests that CAF is widely employed before or during exercise to enhance performance [35,36]. This would mean that it is highly likely that some athletes are habituated to CAF due to daily consumption of caffeine-containing products. The existence of athletes habituated to CAF may be particularly common in sports such as cycling, rowing, triathlon, athletics, and weightlifting, sport disciplines that benefit from the use of ballistic exercises during training to increase power output. Habitual CAF intake modifies physiological responses to acute ingestion of this stimulant by the up-regulation of adenosine receptors [37,38]. This effect would produce a progressive reduction of CAF ergogenicity in those athletes consuming CAF on a regular basis, because the newly created adenosine receptors may bind to adenosine and induce fatigue. However, the fact of habituation to the ergogenic benefits of CAF is still inconclusive. The studies by Dodd et al. [39] and de Souza [40] showed similar responses to endurance exercise after acute CAF intake between low and habitual CAF consumers, although this is not always the case [41]. Considering the above, the use of cross-sectional investigations including participants with different degrees of habituation to CAF may explain the lack of consistency when concluding about tolerance to the ergogenic effect of CAF. Lara’s et al. [42] crossover design showed that the ergogenicity of CAF was reduced when the substance was consumed daily for 20 days, yet afterwards the ergogenic properties of CAF were maintained. On the contrary, tolerance to some of the side-effects associated to CAF has been observed in habitual consumers of CAF [43]. However, only two previous studies analyzed the impact of CAF intake in habitual CAF consumers using resistance exercise test protocols [18,19,20]. These investigations indicate that CAF ergogenicity to power output is mostly reduced in individuals habituated to CAF, while only high doses (>9 mg/kg/b.m.) may exert some benefit in maximal strength [19,20]. However, to date, there is no available data regarding the influence of acute CAF intake on power output and bar velocity during ballistic exercises in athletes habitually consuming CAF.

Given the widespread use of the BPT exercise as a mean of developing power output in the upper limbs [44,45] and the widespread use of CAF in sport, it would be interesting to investigate whether acute CAF intake affects power output and bar velocity in athletes habituated to CAF. For this reason, the aim of the present study was to evaluate the effects of the acute intake of 3 and 6 mg/kg/b.m. of CAF on power output and bar velocity during five sets of the BPT in participants habituated to CAF. It was hypothesized that acute CAF intake would increase power output and bar velocity during the BPT training session when compared to a control situation, even in participants habituated to CAF.

## 2. Materials and Methods

A randomized, double-blind, PLAC-controlled crossover design was used for this investigation. Initially, the participants performed a familiarization session with the experimental protocols that included a 1 RM measurement for the bench press exercise. Afterwards, they performed 3 different experimental sessions with a one-week interval between sessions to allow for complete recovery and a wash-out of ingested substances (Figure 1). During the 3 experimental sessions, the study participants either ingested a PLAC, 3 mg/kg/b.m. of CAF (CAF-3) or 6 mg/kg/b.m. of CAF (CAF-6). One hour after ingestion of CAF or PLAC, they performed 5 sets of 2 repetitions of the BPT exercise at 30% 1 RM. Both CAF and PLAC were administered orally to allow for peak blood CAF concentration during the training session and at least 2 h after the last meal to avoid any interference of the diet with the absorption of the experimental substances. CAF supplementation was provided to participants in the form of unidentifiable capsules (Caffeine Kick^®^, Olimp Laboratories, Debica, Poland). The manufacturer of the CAF capsules also provided identical PLAC capsules filled with all-purpose flour. Participants refrained from strenuous physical activity the day before the experimental trials but they maintained their training routines during the duration of the experiment to avoid any performance decrement due to inactivity. Additionally, the participants maintained their dietary habits during the study period, including daily CAF intake. They received a list of products containing CAF which could not be consumed within 12 h of each experimental trial. Compliance was tested verbally and by using dietary records. Additionally, the participants were required to refrain from alcohol and tobacco, medications or dietary supplements for two weeks prior to the experiment. All subjects registered their calorie intake using the “Myfitness pal” software [46] (Under Armour, Baltimore, MD, USA) every 24 h before the testing procedure, to ensure that the caloric intake was similar between experimental sessions.

### 2.1. Participants

Twelve healthy strength-trained male athletes (age: 25.3 ± 1.7 years., body mass: 88.4 ± 16.5 kg, body mass index (BMI): 26.5 ± 4.7, bench press 1 RM: 128.6 ± 36.0 kg; mean ± SD) volunteered to participate in the study. All participants completed a written consent form after they had been informed of the risks and benefits of the study protocols. The participants had a minimum of 3 years of strength training experience (4.4 ± 1.6 years). All of them were classified as high habitual CAF consumers according to the classification recently proposed by de Souza Gonçalves et al. [40]. They self-reported their daily ingestion of CAF (5.0 ± 0.95 mg/kg/b.m./day, 443 ± 142 mg/day) based on a Food Frequency Questionnaire (FFQ). The inclusion criteria were as follows: (a) free from neuromuscular and musculoskeletal disorders, (b) 1 RM bench press performance with a load of at least 120% body mass, (c) habitual CAF intake in the range of 4–6 mg/day/kg/b.m. The athletes were excluded from the study when they suffered from any pathology or injury or when they were unable to perform the exercise protocol at the maximum effort. The investigation protocols were approved by the Bioethics Committee for Scientific Research at the Academy of Physical Education in Katowice (March 2019), Poland, according to the ethical standards of the latest version of the Declaration of Helsinki, 2013.

### 2.2. Habitual Caffeine Intake Assessment

Daily CAF intake was measured by an adapted version of the Food Frequency Questionnaire (FFQ) proposed by Bühler et al. [47]. Household measures were employed to individually assess the amount of food consumed during a day, week and month. The list was composed of dietary products with moderate-to-high CAF content including different types of coffee, tea, energy drinks, cocoa products, popular beverages, medications, and CAF supplements. Nutritional tables were used for database construction [48,49,50] and an experienced nutritionist calculated the daily CAF intake for each participant.

### 2.3. Familiarization Session and One Repetition Maximum Test

A familiarization session with the experimental procedures preceded 1 RM testing in the bench press exercise. In this session, the athletes arrived at the laboratory between 9:00 and 10:00 am. and cycled on an ergometer for 5 min. Afterwards, they performed 15 repetitions at 20% of their estimated 1 RM in the barbell bench press exercise followed by 10 repetitions at 40% 1 RM, 5 repetitions at 60% 1 RM and 3 repetitions at 80% 1 RM. Then they executed single repetitions of the bench press exercise with a 5 min rest interval between successful attempts. The load for each subsequent attempt was increased by 2.5 to 10 kg, and the process was repeated until failure. Hand placement on the barbell was individually selected grip width (~150% individual bi-acromial distance). After completing the 1 RM test in the bench press exercise, the participants performed a maximal BPT on a Smith machine with a load of 30% 1 RM from 1 RM bench press test result, with a maximal tempo of movement.

### 2.4. Experimental Sessions

During experimental sessions, the athletes participated in three identical training trials. All trials took place between 9.00 and 11.00 am. to avoid the effect of circadian variations on the outcomes of the investigation. After replicating the warm-up procedures of the familiarization trial, the athletes performed 5 sets of the 2 BPT repetitions at 30% 1 RM on the Smith machine. The repetitions were performed without rest to produce a ballistic movement while the rest interval between sets was 3 min. The participants were encouraged to produce maximal velocity during both the eccentric and concentric phase of the BPT movement. Two spotters were present on each side of the bar during the exercise protocol to ensure safety. To standardize the exercise protocol for all trials, each BPT was performed without bouncing the barbell off the chest, with the lower back in contact with the bench and without any pause between the eccentric and concentric phases of the movement. A rotatory encoder (Tendo Power Analyzer, Tendo Sport Machines, Trencin, Slovakia) was used for instantaneous recording of bar velocity during the whole range of motion, as in previous investigations [51]. During each BPT, peak power output (PP, in W) mean power output (MP, in W); peak bar velocity (PV, in m/s); and mean bar velocity (MV, in m/s) were registered. MP and MV were obtained as the mean of the two repetitions while PP and PV were obtained from the best repetition.

### 2.5. Statistical Analysis

Data are presented as the mean ± SD. All variables presented a normal distribution according to the Shapiro-Wilk test. Verification of differences in peak power output (PP), mean power output (MP), peak bar velocity (PV), and mean bar velocity (MV) was performed using a two-way (substance × set) analysis of variance (ANOVA) with repeated measures. In the event of a significant main effect, post-hoc comparisons were conducted using the Tukey’s test. Percent changes and 95% confidence intervals were also calculated. Effect sizes (Cohen’s *d*) were reported where appropriate and interpreted as large (*d* ≥ 0.80); moderate (*d* between 0.79 and 0.50); small (*d* between 0.49 and 0.20); and trivial (*d* < 0.20); [52].

## 3. Results

The two-way repeated measures ANOVA indicated no significant substance × set main interaction effect for MP (F = 1.19; *p =* 0.32); MV (F = 1.18; *p =* 0.32); PP (F = 1.05; *p =* 0.40); PV (F = 1.09; *p =* 0.38). However, there was a significant main effect of substance in MP (F = 7.27; *p* < 0.01) and MV (F = 6.75; *p* < 0.01). No statistically significant main effect of substance was revealed in PP (F = 2.91; *p* = 0.07) and PV (F = 2.63; *p* = 0.09; Table 1).

Post hoc analyses for main effect of substance indicated significant increases in MP (*p* < 0.01; ES = 0.14) and MV (*p* = 0.01; ES = 0.78) in BPT (mean of the 5 sets) after the intake of CAF-3 compared to PLAC as well as significant increases in MP (*p* = 0.01; ES = 0.13) and MV (*p* = 0.01; ES = 0.72) in the BPT (mean of the 5 sets) after the intake of CAF-6 compared to PLAC. There were no significant differences in MP and MV between the two doses of CAF (CAF-3 vs. CAF-6). The results of particular sets in MP, MV, PP, and PV as well ES between PLAC and CAF-3, CAF-6 in each set are presented in Table 2.

Figure 2 and Figure 3 represent the individual responses induced by CAF-3 and CAF-6, in comparison to the placebo, for MP and MV. The 11 out of 12 participants showed an increase in MP and MV after the ingestion of CAF-3, while CAF-6 produced higher values for MP and MV in 10 out of 12 participants.

The y-axis represents the difference in mean bar velocity output during the 5 sets of BPT between PLAC–CAF-3; PLAC–CAF-6 for each individual.

## 4. Discussion

The main finding of this study was that acute CAF intake has a positive effect on MP and MV during a training session of the BPT performed at 30% 1 RM. Interestingly, both 3 and 6 mg/kg/b.m. doses of CAF had similar effectiveness in enhancing performance when compared to PLAC. Additionally, the ergogenic effect of CAF on MP and MV was evident in most of participants, as all of them responded by improving performance with either CAF-3 or CAF-6, even when they were catalogued as individuals habituated to CAF (Figure 2 and Figure 3). However, the study did not show significant changes in PP and PV after CAF intake with either dose of CAF (3 and 6 mg/kg/b.m.) compared to PLAC. These outcomes suggest that acute CAF intake in a moderate dose (from 3 to 6 mg/kg/b.m.) is effective in increasing mean power and bar velocity during the BPT without a significant influence on peak values of these variables. These results suggest that CAF can be effectively used to acutely improve this power-specific training routine even with individuals habituated to CAF, although the long-term training effects with CAF should be further investigated.

The y-axis represents the difference in mean power output during the 5 sets of BPT between PLAC–CAF-3; PLAC–CAF-6 for each individual.

Previous research showed that acute CAF intake increases power output during the bench press exercise [8,14,15,16]. However, most of these studies included only one set of the exercise which is not the habitual practice during sports training, where several sets of a particular exercise are performed in order to obtain significant adaptations derived from training. In presented study, the main effect of increase in MP and MV after the intake of CAF-3 and CAF-6 over the placebo has occurred for training session consisting several sets (Table 2). The ergogenic effect of CAF observed during the BPT is partly consistent with the results of previous findings [8,16]. However, it should be emphasized that this is the first study investigating the effects of CAF during a training session that includes several sets of a ballistic exercise. Experimental procedures with the use of CAF in which more than one set of an upper body exercise are used are scarce [18,53,54]. The study of Lane and Byrd [53] showed that the intake of 300 mg of CAF, representing 3.5 mg/kg/b.m. increased peak velocity during 10 sets of the bench press exercise at 80% 1 RM compared to PLAC. Wilk et al. [18] did not show significant changes after the intake of different doses of CAF (3, 6, and 9 mg/kg/b.m.) in both, mean and peak power output and bar velocity during the BP exercise at 50% 1 RM (3 sets of 5 repetitions), although this investigation was carried out in athletes habituated to CAF. No changes in mean and peak bar velocity after CAF intake of 150 mg, representing 1.74 mg/kg/b.m. were observed in a study by Lane et al. [54] where 10 sets of 3 repetitions of the bench press exercise were performed at 80% 1 RM. The current study is quite innovative because it is the first investigation geared to assess the effects of CAF intake on power output by using a ballistic upper body exercise with a low external load (30% 1 RM), geared for power training of athletes [55]. The current state of the literature, indicates that CAF is useful in increasing power during one or multiple sets of the bench press exercise when the dose ingested is >3 mg/kg/b.m., but it seems particularly effective when using low and moderate loads during an explosive exercise, such as the BPT.

The overall increase of MP and MV during the training session of the BPT after ingestion of CAF-3 and CAF-6 can be also attributed to increased pre-exercise central excitability. Specifically, the pre-exercise ingestion of CAF would allow the athletes to maintain a certain amount of force even in the presence of biochemical changes within the working muscle that lead to fatigue [3]. Under this theory, CAF intake would allow a higher physical performance because it would help to maintain the neural response even in the presence of metabolic perturbations such as low muscle pH. This effect may be accompanied by reductions in interstitial potassium accumulation found after CAF intake [2], that ultimately leads to the maintenance of excitability during exercise [33,56]. In the central nervous system (CNS), CAF binds to adenosine receptors that influence the release of neurotransmitters, such as noradrenaline and acetylcholine [4,57,58,59] and consequently, increase muscle tension [60]. However, in the current investigation, this purported effect of CAF on CNS was not sufficient to enhance PP and PV during the BPT at 30% 1 RM. Thus, reduced fatigue through CAF-induced modulation of both peripheral and central neural processes may explain the obtained results and higher MP and MV of the bar during the BPT training session. Nevertheless, the association of the ergogenic effect with the mechanisms that allowed this ergogenic effect is speculative at this moment because no measurements were carried out to test the origin of caffeine’s ergogenic effects.

It should be taken into consideration that the study participants in the current study were habitual CAF users. In contrast, most of the investigations aimed at determination of the ergogenic effect of CAF on muscle performance have selected individuals unhabituated to this stimulant or with low-to-moderate daily consumption of CAF (e.g., from 58 to 250 mg/day), [11,16,34], to avoid the effects that tolerance to CAF may. However, CAF is an ergogenic aid frequently used in training and competition and it is likely that some athletes seeking for ergogenic benefits of CAF are already habituated to this substance due to the chronic use of caffeine-containing supplements during training and competition. In fact, previous investigations have suggested that between 75% and 90% of athletes use CAF in competitive and training settings [35,36,61], suggesting that studies on the effect of acute CAF intake on physical performance during real training and competition settings are particularly important in athletes habituated to CAF. In this respect, previous research using well-controlled CAF treatments has suggested that the habitual intake of this stimulant may progressively reduce its ergogenic effect on exercise performance [42,62] and then, it has been speculated that the ergogenic effect of CAF could be dampened in habitual CAF users.

To the authors’ knowledge, only three previous studies analyzed power output of the upper limbs in a group of participants habituated to CAF [9,18,19,20]. The study of Sabol et al. [9] showed an increase in medicine ball throwing distance after the acute intake of 6 mg/kg/b.m. of CAF but the doses of 2 and 4 mg/kg/b.m. did not show any differences with the PLAC. The study by Wilk et al. [18] did not show increases in power output and bar velocity during the bench press exercise in high habitual CAF users that ingested from 3 to 9 mg/kg/b.m. Although it has been theorized that the reduction in the ergogenic effects of CAF in habitual users can be modified using doses greater than the daily habitual intake [63], previous investigations indicate that athletes habituated to CAF do not benefit from the acute ingestion of CAF in doses above their habitual intake while the prevalence of side effects is greatly increased [19,20]. Interestingly, participants in the presented study self-reported their daily ingestion of CAF, which amounted to 5.0 ± 0.95 mg/kg/b.m., (443 ± 142 mg of CAF per day), and the acute CAF doses (especially CAF-3) and some performance enhancements were obtained even when de dose of CAF did not exceed the value of habitual consumption. In any case, although the current investigation found a positive effect of CAF on mean power output and mean bar velocity during the BPT in athletes habituated to CAF, it is still possible that the effect of this substance is higher in unhabituated individuals.

In addition to its strengths, the current study presents limitations that should be addressed. Although the results showed a significant main effect on MP and MV after CAF intake, the direct causes of these changes cannot be determined and explained. The study did not include biochemical analysis which could explain the obtained results. In addition, blood samples were not obtained and thus, we have no data about serum CAF concentrations with each of the dosages of CAF employed in this investigation. Further, we did not analyze the genetic intolerance on CAF in the tested subjects. However, the participants of this study did not report any side effects after consuming CAF in the six months prior to the experiment. Due to the fact that the response to CAF is related to the individual tolerance of this substance [42], the dose [19,20], and gender [64] therefore the results of this study should only be translated to males habituated to CAF who use low to moderate CAF doses to enhance performance. Another limitation of the study was that the 1 RM test was performed using the barbell bench press exercise while the BPT was performed on a Smith machine during the experimental trials to increase the security of participants and investigators. Although there is a high transfer between the results obtained in both types of exercise, the calculation of loading would be more reliable if both evaluations were performed on the same resistance exercise. In any case, this limitation did not affect the outcomes of the investigation because the load was the same for all experimental trials.

## 5. Conclusions

The results of the present study indicate that acute doses of CAF, between 3 and 6 mg/kg/b.m., ingested before the onset of an explosive resistance exercise produced an overall effect on mean power output and mean bar velocity during a BPT training session in a group of habitual CAF users. The main effect in mean power and bar velocity was found in several sets during the trial which may indicate that the use of CAF was effective in increasing performance in the whole training session. In contrast, no significant changes were observed for peak power output and peak bar velocity. These results suggest that the ingestion of CAF prior to ballistic exercise can enhance the outcomes of resistance training. However, the results of our study refer only to power output and bar velocity of the upper limbs during the BPT with an external load of 30% 1 RM and further investigations should consider the effect of CAF with different loads or the use of lower-body exercises.

## Figures and Tables

**Figure 1 nutrients-12-00406-f001:**
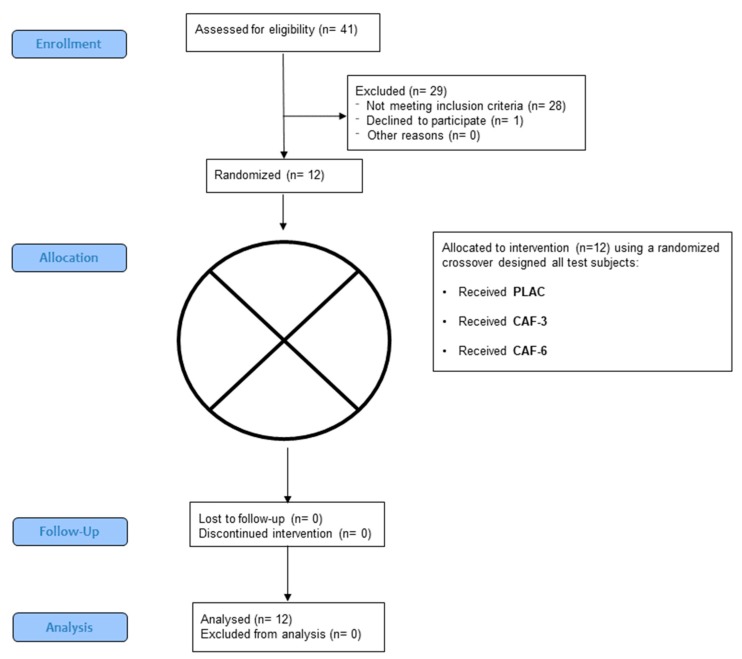
CONSORT flow diagram. n—number of participants; PLAC—placebo; CAF-3—caffeine 3mg/kg/b.m; CAF-6—caffeine 6mg/kg/b.m.

**Figure 2 nutrients-12-00406-f002:**
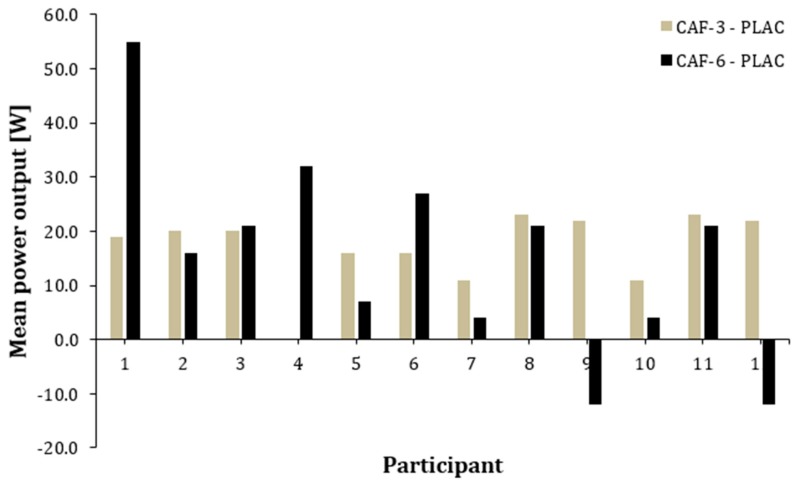
Individual differences in mean power output during 5 sets of bench press throw (BPT) between caffeine and placebo conditions.

**Figure 3 nutrients-12-00406-f003:**
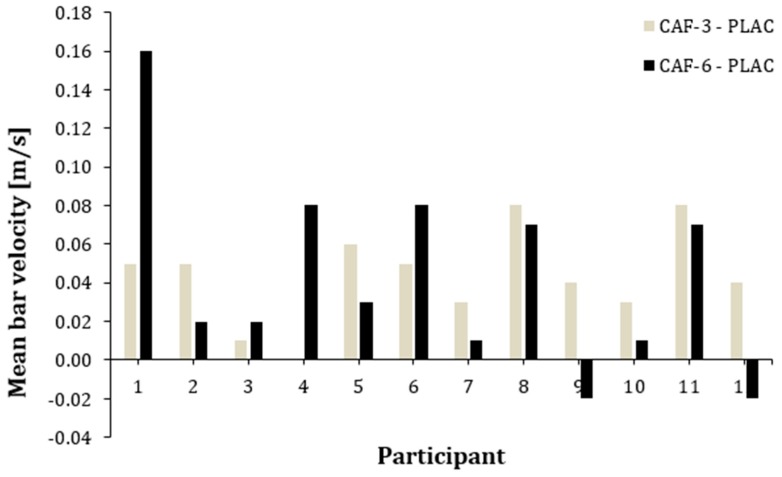
Individual differences in mean bar velocity during 5 sets of bench press throw (BPT) between caffeine and placebo conditions.

**Table 1 nutrients-12-00406-t001:** The main effect for substance on performance variables measured during 5 sets of the bench press throw with the ingestion of 3 and 6 mg/kg/b.m. of caffeine or a placebo.

Bench Press Throw (Mean of the 5 Sets)	Conditions			*p*
	PLAC	CAF-3	CAF-6	
Mean Power (W)	545 ± 117	562 ± 118	560 ± 107	0.01 *
Peak Power (W)	1250 ± 274	1261 ± 220	1297 ± 293	0.07
Mean Velocity (m/s)	1.37 ± 0.05	1.41 ± 0.05	1.41 ± 0.06	0.01 *
Peak Velocity (m/s)	2.14 ± 0.10	2.16 ± 0.07	2.17 ± 0.13	0.09

These data represent the mean values of the 5 sets. All data are presented as mean ± standard deviation. * significant main substance effect. PLAC: placebo; CAF-3: caffeine 3mg/kg/b.m; CAF-6: caffeine 6mg/kg/b.m.

**Table 2 nutrients-12-00406-t002:** Power output and bar velocity during 5 sets of the bench press throw with the ingestion of 3 and 6 mg/kg/b.m. of caffeine or with a placebo.

Conditions		Set 1	Set 2	Set 3	Set 4	Set 5
Mean Power (W)						
PLAC		542 ± 126	548 ± 119	551 ± 117	543 ± 115	540 ± 114
(95%CI)		(462 to 622)	(472 to 623)	(477 to 626)	(470 to 616)	(468 to 613)
CAF-3		552 ± 124	564 ± 115	567 ± 116	557 ± 112	570 ± 124
(95%CI)		(473 to 631)	(490 to 637)	(493 to 640)	(486 to 628)	(492 to 649)
CAF-6		559 ± 109	563 ± 107	562 ± 113	556 ± 103	562 ± 105
(95%CI)		(489 to 628)	(495 to 631)	(489 to 634)	(491 to 621)	(496 to 629)
ES	PLAC vs. CAF-3	0.07	0.14	0.14	0.12	0.25
	PLAC vs. CAF-6	0.14	0.13	0.09	0.12	0.20
**Peak Power (W)**						
PLAC		1245 ± 248	1252 ± 291	1286 ± 378	1244 ± 252	1222 ± 250
(95%CI)		(1088 to 1402)	(1067 to 1437)	(1045 to 1526)	(1083 to 1404)	(1063 to 1381)
CAF-3		1243 ± 218	1252 ± 265	1285 ± 216	1241 ± 199	1283 ± 224
(95%CI)		(1105 to 1382)	(1084 to 1420)	(1147 to 1422)	(1114 to 1368)	(1141 to 1425)
CAF-6		1253 ± 294	1338 ±362	1338 ± 344	1278 ± 255	1278 ± 250
(95%CI)		(1066 to 1440)	(1107 to 1568)	(1119 to 1556)	(1116 to 1440)	(1119 to 1437)
ES	PLAC vs. CAF-3	0.01	0.00	0.01	0.01	0.26
	PLAC vs. CAF-6	0.03	0.26	0.14	0.13	0.22
**Mean Velocity (m/s)**						
PLAC		1.36 ± 0.06	1.38 ± 0.08	1.38 ± 0.05	1.37 ± 0.04	1.36 ± 0.07
(95%CI)		(1.32 to 1.40)	(1.33 to 1.43)	(1.35 to 1.41)	(1.35 to 1.41)	(1.32 to 1.40)
CAF-3		1.39 ± 0.07	1.42 ± 0.05	1.43 ± 0.05	1.40 ± 0.05	1.43 ± 0.05
(95%CI)		(1.34 to 1.43)	(1.39 to 1.45)	(1.40 to 1.46)	(1.37 to 1.43)	(1.40 to 1.46)
CAF-6		1.41 ± 0.07	1.42 ± 0.07	1.41 ± 0.05	1.40 ± 0.07	1.42 ± 0.09
(95%CI)		(1.36 to 1.45)	(1.37 to 1.47)	(1.38 to 1.45)	(1.35 to 1.44)	(1.37 to 1.47)
ES	PLAC vs. CAF-3	0.46	0.60	1.0	0.40	1.15
	PLAC vs. CAF-6	0.77	0.60	0.6	0.33	0.77
**Peak Velocity (m/s)**						
PLAC		2.13 ± 0.08	2.15 ± 0.11	2.17 ± 0.13	2.15 ± 0.12	2.10 ± 0.14
(95%CI)		(2.08 to 2.18)	(2.08 to 2.22)	(2.09 to 2.26)	(2.07 to 2.22)	(2.01 to 2.19)
CAF-3		2.14 ± 0.08	2.17 ± 0.07	2.18 ± 0.09	2.14 ± 0.05	2.19 ± 0.08
(95%CI)		(2.09 to 2.18)	(2.12 to 2.22)	(2.12 to 2.24)	(2.11 to 2.17)	(2.14 to 2.25)
CAF-6		2.16 ± 0.12	2.18 ± 0.13	2.19 ± 0.14	2.17 ± 0.14	2.17 ± 0.17
(95%CI)		(2.08 to 2.23)	(2.10 to 2.26)	(2.10 to 2.28)	(2.09 to 2.26)	(2.06 to 2.28)
ES	PLAC vs. CAF-3	0.13	0.22	0.09	0.11	0.79
	PLAC vs. CAF-6	0.29	0.25	0.15	0.15	0.45

All data are presented as mean ± standard deviation. CI: confidence interval. ES: effect size.

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
