# Peer review of "Acute Caffeine Intake Enhances Mean Power Output and Bar Velocity during the Bench Press Throw in Athletes Habituated to Caffeine"

_nutrients, 2020, doi:10.3390/nu12020406_

Round 1

Reviewer 1 Report

This study aims to demonstrate the effects of caffeine supplementation on bench press throw performance in athletes habituated to caffeine. This is an important topic regarding various different sports disciplines with different performance characteristics. However, the reviewer has some concerns. The detailed comments are presented below.

Introduction:

- In the introduction the authors need to be more objective. It took them to get to the specific topic of the study. The hypothesis proposed are not clearly enough. It would be of interest to better explain the benefits and improvements of measure changes in physical performance with this type of exercise (bench press throw) and supplement. It is necessary to justify better the present study. Also, the effects of caffeine in power performance and its benefits must be included in introduction section. Finally, the study used athletes habituated to caffeine and this reviewer thinks that it is a very interesting point in the present research. Therefore, It is necessary to develop more in detail the rationale about caffeine supplementation and habituated caffeine athletes.  

Methods:

-Did you have any dropouts and could you add the CONSORT diagram of the data collection process?

- I am concerned that the statistical analyses have been misinterpreted. The authors stated that they employed a repeated measures ANOVA with Tukey post hoc, which I believe is suitable for their study design. However, the authors did not report the results accordingly. That being said, interaction effects between set and substance should be reported first (F, ES and p), followed by the main effects or simple main effects for substance or set, then post hoc comparisons. Therefore, It is now apparent in results section that there were no significant interaction effects (substance * set). With no significant interaction, it is not appropriate to interpret post hoc results for the interaction. You can only report on main effects for set or main effects for substance, but not how Placebo and Caffeine had different results to each other at a specific set- this is based on the post hoc for the interaction. Unfortunately, this would mean that most of the discussion which focuses on between-substances differences (i.e. the interaction effects) is redundant, and the paper would have to be largely re-written.

Results:

-Were there any inter-individual differences for the amount of benefit? Could you show some individual data.

Discussion:

There are explanations in the discussion that are not clearly explained

Hypotheses are developed during the discussion that have not been demonstrated with the results of the study. I think the discussion is very speculative and flawed. All the discussion is based on variables that the authors did not measure.

-More discussion about the point that the sample was “athletes habituated to caffeine” would be interesting.

-missing results for plasma caffeine, epinephrine and norepinephrine concentration is a limitation of the study and needs to be mentioned.

Author Response

Thank you for the time and effort employed for reviewing this manuscript. We have addressed all the points raised by the Reviewer in this response letter and we have made changes where appropriate in the manuscript and have highlighted our responses in red. We really appreciate your time, and we hope that our revision has satisfactorily met your expectations.

Introduction:

- In the introduction the authors need to be more objective. It took them to get to the specific topic of the study. The hypothesis proposed are not clearly enough. It would be of interest to better explain the benefits and improvements of measure changes in physical performance with this type of exercise (bench press throw) and supplement. It is necessary to justify better the present study. Also, the effects of caffeine in power performance and its benefits must be included in introduction section. Finally, the study used athletes habituated to caffeine and this reviewer thinks that it is a very interesting point in the present research. Therefore, It is necessary to develop more in detail the rationale about caffeine supplementation and habituated caffeine athletes.  

Reply – Thank you this comment. Following your suggestions, we have updated the introduction to show the benefits that acute caffeine intake might produce in this type of exercise and we have included a new introductory paragraph about the use of athletes habituated to caffeine. Finally, we have enhanced the study hypothesis.

P 2, L 61-69; P 2-3 – L 83-109; P 3 – 110-117

Methods:

-Did you have any dropouts and could you add the CONSORT diagram of the data collection process?

Reply – Thanks for this suggestion. We have included a new Figure 1 with the CONSORT diagram.

P 4

- I am concerned that the statistical analyses have been misinterpreted. The authors stated that they employed a repeated measures ANOVA with Tukey post hoc, which I believe is suitable for their study design. However, the authors did not report the results accordingly. That being said, interaction effects between set and substance should be reported first (F, ES and p), followed by the main effects or simple main effects for substance or set, then post hoc comparisons.

Reply – Thanks for this insightful comment. We have used your suggestion about the presentation of the results as “(F, ES and p)”. Regarding the interpretation of the result, it is true that some researchers do not use post-hoc analysis in absence of interaction effects in those investigations using two-ways ANOVAS of repeated measures. However, previous investigations have suggested that there is a misconception of statistics when avoiding post-hoc analysis in absence of interactions (please, see Wei et al, 2012). In fact, if there is a main effect for only one factor, multiple comparisons among the treatment means for this factor are required to identify which specific means differ. Following this indication, we have performed post-hoc analysis only in those variables where a main effect of caffeine was found.

Wei et al. Comparisons of treatment means when factors do not interact in two-factorial studies. Amino Acids. 2012 May;42(5):2031-5. doi: 10.1007/s00726-011-0924-0.

P 5, L 204-214

Therefore, It is now apparent in results section that there were no significant interaction effects (substance * set). With no significant interaction, it is not appropriate to interpret post hoc results for the interaction. You can only report on main effects for set or main effects for substance, but not how Placebo and Caffeine had different results to each other at a specific set- this is based on the post hoc for the interaction. Unfortunately, this would mean that most of the discussion which focuses on between-substances differences (i.e. the interaction effects) is redundant, and the paper would have to be largely re-written.

Reply - Again, thanks for this suggestion. As we have indicated in the previous comment, we are now only employing CAF-PLAC post hoc comparison to maintain this analysis within the main effect found in MP and MV.

Unfortunately, this would mean that most of the discussion which focuses on between-substances differences (i.e. the interaction effects) is redundant, and the paper would have to be largely re-written.

Reply - Thanks for this comment. We have now discussed the main effect and the PLAC-CAF comparisons derived from the post-hoc analysis.

Results:

-Were there any inter-individual differences for the amount of benefit? Could you show some individual data.

Reply – Thanks for this comment. Following your suggestion, we have included two figures that show the individual response for mean power and mean bar velocity.  

P – 7

Discussion:

There are explanations in the discussion that are not clearly explained.

Reply – Thanks for this comment. We have modified several parts of the discussion to improve its readability.

Hypotheses are developed during the discussion that have not been demonstrated with the results of the study. I think the discussion is very speculative and flawed. All the discussion is based on variables that the authors did not measure.

Reply – Thanks for this comment. We have deleted those parts that might be considered speculative.

-More discussion about the point that the sample was “athletes habituated to caffeine” would be interesting.

Reply – Thanks for this comment. We have enlarged our discussion about the fact that the athletes were habituated to caffeine due to chronic consumption.

-missing results for plasma caffeine, epinephrine and norepinephrine concentration is a limitation of the study and needs to be mentioned.

Reply – Thanks for this comment. The expert Reviewer is correct. We have included this as a limitation of the investigation.  

P 9, L 327- 330

On behalf of all co-authors, many thanks for the insightful comments and suggestions for this review.

Reviewer 2 Report

Thank you for providing me the chance to review this manuscript. This is a novel approach to measuring the effects of caffeine intake on athletes. The novelty is the BPT. The manuscript is well written and concise in each section. Below are a few minor additions to increase the strength of the manuscript.

Title: Acute Effect of Caffeine Ingestion on Power Output and Bar Velocity During the Bench Press Throw in Athletes Habituated to Caffeine

Title: Title is appropriate and represents the article. It does not represent the results though.

Keywords: You might want to include caffeine in the keywords

Intro: Provided a good background of current literature and a justification for the study. Suggest adding habitual CAF consumers in the hypothesis.

Methods:

Line 92: Do you mean 60 mins after CAF or PLAC and not just “CAF”?

Line 105: Was the diet similar or not statistically different between the trials? Also, were there variations in the macros? What about foods that contain caffeine 12 hours before?

Line 136: Was the 1RM done on a Smith Machine?

Results:

Table 1: It may help to add a footnote to define the acronyms. This would help the table stand alone.

The tables were appropriate.

Discussion: Well written and concise.

Author Response

Reply – Thank you for the time and effort employed for reviewing this manuscript. We have addressed all the points raised by the Reviewer in this response letter and we have made changes where appropriate in the manuscript and have highlighted our responses in red. We really appreciate your time, and we hope that our revision has satisfactorily met your expectations.

Title: Acute Effect of Caffeine Ingestion on Power Output and Bar Velocity During the Bench Press Throw in Athletes Habituated to Caffeine

Title: Title is appropriate and represents the article. It does not represent the results though.

Reply – Thanks for this suggestion. We have modified the title to show the direction of the results.

Keywords: You might want to include caffeine in the keywords

Reply - The word “caffeine” is in the title and since the guidelines for authors indicate that the words contained in the title should not repeat in keywords, we cannot make such a change.

Intro: Provided a good background of current literature and a justification for the study. Suggest adding habitual CAF consumers in the hypothesis.

Reply – Thanks for this comment. We have included in the hypothesis the fact that the individuals were habitual caffeine consumers.

P 3, L 110-117

Methods:

Line 92: Do you mean 60 mins after CAF or PLAC and not just “CAF”?

Reply - of course after CAF and PLAC. The change has been made.

P 3, L 125

Line 105: Was the diet similar or not statistically different between the trials? Also, were there variations in the macros? What about foods that contain caffeine 12 hours before?

Reply – The diet in the last 24h was controlled and it was not different in kcal nor in the proportion of macros. Participants were instructed to avoid caffeine-containing products and they were provided with a list of products containing caffeine that could be not consumed 12h before the test. According to this suggestion this information was added

P 3, L 133-140

Line 136: Was the 1RM done on a Smith Machine?

Reply – due to safety and habits of participants, the 1RM was made in barbell bench press. This information was added to method section as well to limitation of study. In any case, this limitation did not affect the outcomes of the investigation because the load was the same for all experimental trials.

P3, L 121

P4, L 167

P 4, L 169-170

Results:

Table 1: It may help to add a footnote to define the acronyms. This would help the table stand alone.

Reply – changes have been made

The tables were appropriate.

Discussion: Well written and concise.

Reply – Thanks for this comment.

On behalf of all co-authors, many thanks for the insightful comments and suggestions for this review.

Round 2

Reviewer 1 Report

The authors have replied all the comments